# PVDF/KNO_3_ Composite Sub-Microfibers Produced by Solution Blow Spinning as a Hydrophobic Matrix for Fertilizer Delivery System

**DOI:** 10.3390/polym14051000

**Published:** 2022-03-01

**Authors:** Fabio Sobral, Michael J. Silva, Thalita Canassa, Além-Mar Goncalves, Cícero Cena

**Affiliations:** 1Programa de Pós-Graduação em Química, Instituto de Química, UFMS—Universidade Federal de Mato Grosso do Sul, Campo Grande 79070-900, Brazil; fsnogueira@hotmail.com (F.S.); thalita.canassa@ufms.br (T.C.); alem-mar.goncalves@ufms.br (A.-M.G.); 2UNESP—Universidade Estadual Paulista “Júlio de Mesquita Filho”, Rosana 19272-000, Brazil; michael.silva@unesp.br; 3Programa de Pós-Graduação em Ciência dos Materiais, Instituto de Física, UFMS—Universidade Federal de Mato Grosso do Sul, Campo Grande 79070-900, Brazil

**Keywords:** nanofiber, controlled release, solution spinning, PVDF

## Abstract

Nutrient supplementation is a common practice in agriculture to increase crop productivity in the field. This supplementation is usually excessive, causing nutrient leaching in periods of rainfall leading to environmental problems. To overcome such issues, many studies have been devoted to developing polymeric matrices for the controlled and continuous release of nutrients, reducing losses, and keeping plants nourished for as long as possible. However, the release mechanism of these matrices is based on water diffusion. They start immediately for swellable polymeric matrices, which is not interesting and also may cause some waste, because the plant only needs nutrition only after the germination process. Here, as proof of concept, we tested a hydrophobic polymeric matrix based on sub-microfibers mats, produced by solution blow spinning, filled with potassium nitrate (KNO_3_) for the controlled release of nutrients to plants. In this work, we used the polyvinylidene fluoride (PVDF) polymer to produce composite nanofibers containing pure potassium nitrate in the proportion of 10% weight. PVDF/KNO sub-microfibers mats were obtained with 370 nm average diameter and high occurrence of beads. We performed a release test using PVDF/KNO_3_ mats in a water bath. The release kinetic tests showed an anomalous delivery mechanism, but the composite polymeric fibrous mat showed itself to be a promising alternative to delay the nutrient delivery for the plants.

## 1. Introduction

Polymeric nanofibers exhibit a high surface area and offer several possibilities for applications. The easy incorporation of active ingredients into polymeric fibers motivates their use as controlled delivery systems for agriculture applications once they provide a regular and continuous supply of agrochemicals to soil and/or plants, reducing eventual losses and environmental pollution [1].

Fibrous mats produced by electrospinning techniques have been used for different proposed applications on agricultural activities, such as controlled delivery of pheromones in traps for pest control [2,3,4], fungicide protection for rice seeds [5], allowing temperature control, water and air exchange with the surrounding environment, and the improvement of the soil microbiome by releasing encapsulated microbes and bacteria [6,7], since they protect the microorganism from environmental stress, such as high temperature and dehydration.

Fertilizer delivery systems can also be produced, and their release rate can be tuned by changing the polymeric matrix and fibrous mat microstructure [8,9]. Additionally, fiber mats prevent fertilizer leaching due to rain, which reduces the amount of fertilizer introduced in the soil for the plant, avoiding pollution of river networks. During the dry season, the fiber mats aid seed fixation in the soil [10]. Finally, fibrous mats can be used as a removal material for pollutants, such as pesticides and herbicides, from soil and water [11,12].

An extensive range of materials has been produced as nanofibers by the electrospinning technique [13]. This method exhibits an expensive experimental setup and lower production rates making it a disadvantageous approach for large-scale production. On the other side, the solution blow spinning [14,15] can be scalable (showing a production rate 10 times higher) and requires an inexpensive setup, which permits its use directly in the field. Solution blow spinning microfibers are extensively studied, and polymeric, composite, and ceramic fibers have been produced. The influence of experimental parameters on fiber mats microstructure is well described in the literature [16,17,18].

Poly (vinylidene fluoride) (PVDF) is a semicrystalline polymer widely studied due to its piezoelectric properties and its ability to be chemically and physically stable. It can be obtained in different phases: orthorhombic alpha (α), beta (β), and delta (δ) phases and the monoclinic gamma (γ) phase [19,20,21]. PVDF has high hydrophobicity, a factor that can make specific applications difficult, as it directly influences the absorption of the material [22].

The controlled release mechanism of agrochemicals can be described by Fick’s First Law of Diffusion, where the diffusion coefficients and the coating thickness play an important role in the release rate [1]. The solute diffusion and water permeation rate through the polymeric matrix are important parameters for tuning the release of agrochemicals. The choice of a proper polymeric matrix is usually based on environmentally friendly materials, which are usually highly permeable for water molecules. In this study, we produced PVDF sub-microfiber mats, loaded with KNO_3_ fertilizer by solution blow spinning technique. We tested the microfiber mats as a hydrophobic matrix for a KNO_3_ delivery system.

## 2. Materials and Methods

### 2.1. Sample Preparation

Poly (vinylidene fluoride) (PVDF), (C_2_H_2_F_2_)_n_ was supplied in powder form from Kynar^®^ 763. The PVDF was dissolved in *N*,*N*-dimethylformamide (DMF), (CH_3_)_2_NC(O)H, supplied from Synth, under constant stirring for 2 h at room temperature (25 °C); after complete dissolution, acetone (AC), C_3_H_6_O, supplied from Dinâmica, was added in 20 vol% to improve the solvent volatility a few minutes before the sample production by solution blow spinning.

Polymeric nanofibers were obtained by solution blow-spinning technique, with the aid of an airbrush system, as described elsewhere [23,24]. The best conditions for polymeric fiber formation were determined by studying the influence of the DMF/acetone ratio and gas pressure on fiber morphology. The best conditions found for PVDF fiber production were kept for future composite fiber deposition. The standard processing condition for nanofiber formation was a gas pressure of 60 Psi, a work distance of 25 cm, and a 22G needle. We produced PVDF microfibers, loaded with KNO_3_ as fertilizer, by adding 10 wt% of potassium nitrates (Quimex Industry, Midlothian, IL, USA) into the PVDF solution before fiber processing.

After adjusting to the best conditions for fiber production, the PVDF and PVDF/KNO_3_ were continuously produced for 1 h in a flow hood chamber, then the fiber mats were collected, Figure 1, for future tests and characterization. 

### 2.2. Sample Characterization

SEM (Scanning Electron Microscopy) images were obtained in a JEOL, model JSM-6380LV, with 8–10 kV, after sample surface metallization with gold thin film by a sputtering technique. The fibers’ diameters were measured using IMAGE J software (National Institutes of Health, Wayne Rasband, Bethesda, MD, USA) by analyzing 100 random fibers from each image; a histogram was produced for each sample and the higher-occurrence fiber diameters were used as a reference for describing each sample.

Attenuated total reflectance Fourier transform infrared spectroscopy (ATR-FTIR), performed in a Perkin-Elmer Spectrum 100 model, investigated the presence of KNO_3_ in the PVDF sub-micrometer fibers. The measurements ranged from 4000 to 600 cm^−1^ with 4 cm^−1^ resolution and 5 scans; the samples were measured in three different regions and the average FTIR spectra were used as reference. Thermogravimetric analysis (TGA) was performed in Shimadzu TGA-50 equipment, under a nitrogen atmosphere with 60 mL/min steady flow, from room temperature up to 800 °C, to evaluate the thermal stability of the samples.

The fiber mats’ wettability was estimated by contact angle measurement in a contact angle analyzer Tantec, CAM-Micro model, by sessile drop technique. The sessile drop method is the most used method for determining the wettability of a solid surface. It consists of depositing a drop of a known liquid on a solid surface and measuring the angle formed between the drop and the solid. The contact angle (θ) value can vary from 0° to 180°. A θ = 0° is defined as complete wettability (hydrophilic). On the other hand, when θ = 180° there is no wettability (hydrophobic). The fiber mats were carefully deposited onto a glass substrate to avoid surface folding during the measurements; the tests were made in triplicate by monitoring the contact angle of a deionized water drop on the fiber mat surface for 80 s. The average value used as reference. Finally, the KNO_3_ kinetic release was measured by introducing the PVDF/KNO_3_ fiber mats in a deionized water bath with a weak constant stirring. The solution conductivity was measured by a portable conductometer to estimate the ionic concentration in the solution. Kinetic release models were investigated to describe and understand the delivery mechanism of the PVDF sub-micrometer fibers.

## 3. Results and Discussion

The best conditions for PVDF sub-microfiber production were investigated using different solution parameters (DMF/acetone ratio), and by varying the gas pressure in the airbrush. The only conditions that provided enough sample formation for the tests are shown in Figure 1. Small gas pressure was not able to pull out the fibers and drag them to the collector for a solution with concentrations of 0.12 g/mL in our experimental setup; sample formation was only possible above 40 Psi.

The samples produced with highly volatile solution, by introducing acetone above 20% in volume, Figure 2a–c, did not show a pure fiber formation. Figure 2a shows two different classic formations, the first not homogenous. Thick and flattened fiber formation can be observed because the solvent volatility and drag forces can pull fibers from the tip, but the high solvent volatility causes the uneven ejection of more material. In the second, entire drops are ejected from the tip and the exploded drop morphology is also observed on the fiber mat.

In Figure 2b, smooth, uniform, and thin fiber formation are observed in low amounts, but several entire drops were obtained. The air flux causes fast solvent evaporation, then entire dry drops can be ejected from the tip, and several defects in the fiber mat increase; similar behavior has been reported in the literature [14]. By adjusting the solution volatility, the fiber formation started (Figure 2c) but nonetheless a lot of defects were observed due to the entire drop ejection from the tip, following which exploded drop formation could be observed. Finally, smooth and uniform fiber formation was obtained after fine adjustments in the experimental parameters (gas pressure, injection rate, work distance) improving the fiber quality (Figure 2d). 

Figure 3 shows SEM images and histograms of the obtained sub-microfibers produced from pure PVDF and PVDF/KNO_3_ solution. The respective histograms showed a monomodal distribution, with 285 nm and 370 nm fiber average diameter. PVDF sub-microfibers showed a smooth and continuous morphology (Figure 3a). The introduction of KNO_3_ into the solution may promote the alteration of the solution’s rheological properties. As a result, we can observe a high concentration of bead formation in the mat (yellow circle), Figure 3b. 

The thermal degradation for both samples starts only above 370 °C, which does not influence the desirable process for this type of application (Appendix A). The FTIR spectra for PVDF and PVDF/KNO_3_ sub-microfibers are presented in Figure 4a. Dissimilar bands are evidenced between 1800 cm^−1^ and 1070 cm^−1^. This happens, probably, because of changes in the proportion of PVDF phases in the fibers promoted by introducing KNO_3_ in the solution. Nonetheless, the main band assigned to NO_3_^−^ symmetric vibration around 1363 cm^−1^ was not evidenced [25]. Asymmetric axial deformation of CH_2_ (2923 and 2852 cm^−1^); axial deformation due to C=O (1739 and 1710 cm^−1^), followed by C=C (1672 cm^−1^); CH bending (1465 and 1432 cm^−1^); and a strong band at 1402 cm^−1^ assigned CH_2_ axial deformation was observed only in PVDF spectra. A wide vibrational band starts at 1272 cm^−1^ (C-C axial and angular deformation) and goes until 1157 cm^−1^ in PVDF fibers assigned to C-N and CF_2_ vibration, and appears more detailed for PVDF/KNO_3_. Finally, we observe the CF_2_ angular deformation bands (1070, 872, and 760 cm^−1^) with CH_2_ angular deformation (838 cm^−1^) are quite different for both samples [23,24].

PVDF is a polymer with extremely low surface energy, which guarantees high contact angles observed in Figure 4b, demonstrating its hydrophobicity [26,27]. The contact angles of PVDF and PVDF/KNO_3_ were approximately 120° and 125°, respectively, and they did not change over time. Usually, in practice, a dynamic phenomenon of contact angle hysteresis is often observed because the liquid start to evaporate and should maintain the contact area, but there are two pure modes of evaporation of liquid drops on surfaces, one at the constant contact area and one at constant contact angle [28].

The release test performed for PVDF/KNO_3_ sub-microfibers is shown in Figure 5. The release mechanism for swellable polymeric matrices is a complex phenomenon. The mechanism can be classified as purely diffusion or erosion, or by the combination of both. Fick’s equation is the most common law to describe the model, in which the exponent “n” indicates the release mechanism (n ≤ 0.45 for diffusion-controlled mechanism—case I; n = 0.89 for case II, a relaxational release; non-Fickian, zero-order, is found for n > 0.89—values between 0.45 and 0.89 can be assigned to both phenomena) [29,30,31]. The release mechanism found for PVDF/KNO_3_ sub-microfibers does not obey Fick’s law, as demonstrated in Figure 5 by the orange dashed line, with an R^2^ of 0.82 and n parameter of 0.34. Other models were also tested such as Korsmeyer–Peppas and Higuchi, but failed to fit the data [30,31]. The best fit was obtained for our results by the empirical exponential equation y=ea+bx+c, with R^2^ of 0.92, whose terms had no specific meaning. We add a Langmuir fit, with an R^2^ of 0.93, just to explore the form of an equation that can perhaps better describe the data.

Our attempt to describe the model was unsuccessful, suggesting the anomalous behavior of our polymeric fibrous mat for fertilizer delivery. The data show a prominent release at the first 48 h. Subsequently, the release continues almost stably, which is very interesting but does not solve completely the problem for polymeric matrix delivery systems. The ideal way would be to observe a smaller delivery for 48 h. However, it is good enough in comparison with current models of direct fertilizing application in the soil that reach a leaching loss of almost 100% in the first 24 h.

## 4. Conclusions

PVDF/KNO_3_ sub-microfibers were successfully obtained by solution blow spinning, exhibiting many beads. The composite fiber mats showed a considerable difference in FTIR spectra, probably related to changes in the proportions between the different phases exhibited by the PVDF polymeric chain. The hydrophobic nature of the material was preserved by adding KNO_3_ into the matrix. Kinetic release tests reveal that PVDF/KNO_3_ sub-microfibers do not obey Fick’s law. Nonetheless, their release behavior cannot avoid the fertilizer delivery in the first 48 h, before the seed germination. New studies of composite and blend matrices can drive us to interesting delivery systems for agriculture.

## Figures and Tables

**Figure 1 polymers-14-01000-f001:**
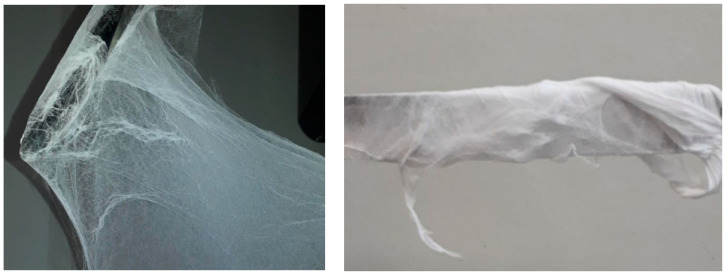
PVDF sub-microfiber mat produced by solution blow spinning.

**Figure 2 polymers-14-01000-f002:**
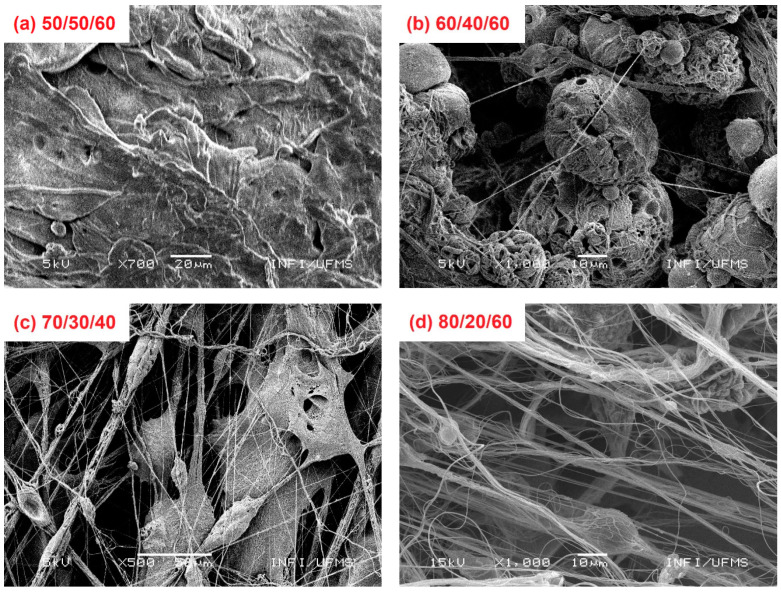
Sample formation for different experimental conditions. (**a**) DMF/acetone = 50/50, gas pressure 60 Psi; (**b**) DMF/acetone = 60/40, gas pressure 60 Psi; (**c**) DMF/acetone = 70/30, gas pressure 40 Psi; (**d**) DMF/acetone = 80/20, gas pressure 60 Psi.

**Figure 3 polymers-14-01000-f003:**
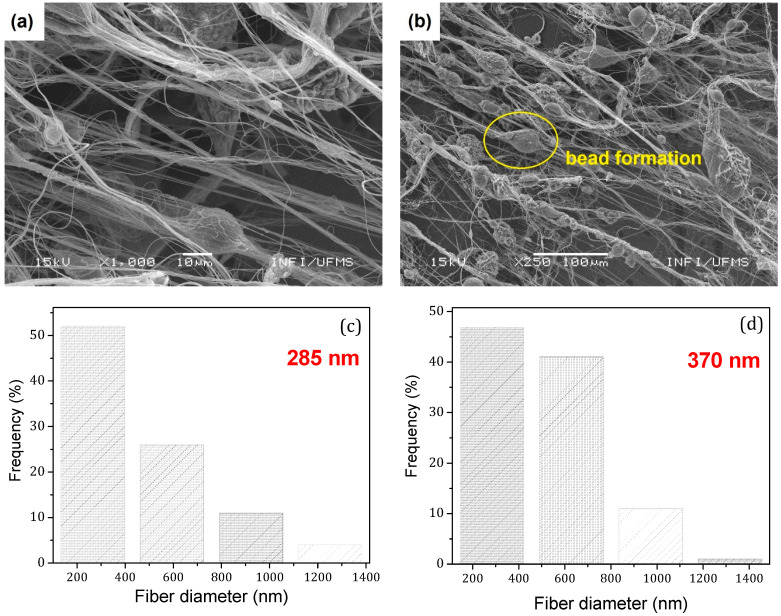
SEM image (**a**) of PVDF fibers (0.1 g/mL) and (**b**) PVDF/KNO_3_ fibers; (**c**,**d**) are the respective histograms, with average diameter values around 285 nm and 370 nm, respectively. The bead formation is evidenced by the round yellow mark in (**b**).

**Figure 4 polymers-14-01000-f004:**
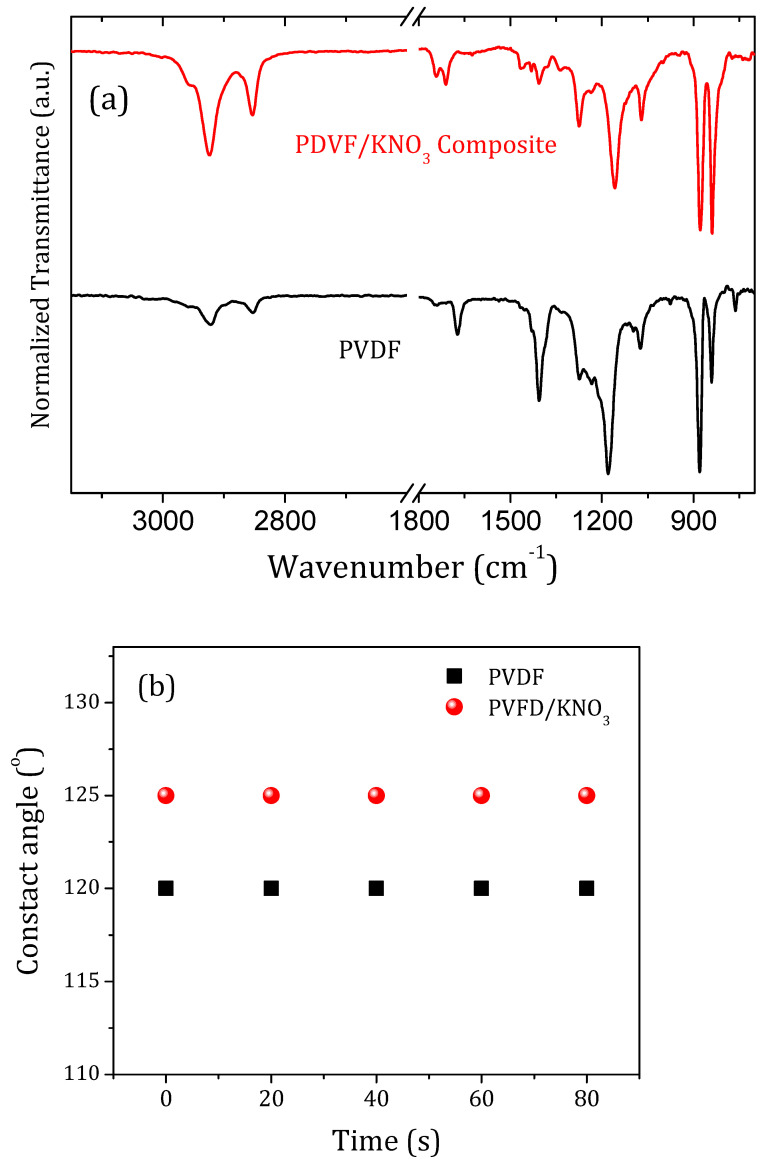
(**a**) FTIR spectra and (**b**) contact angle for PDVF and PVDF/KNO_3_ sub-microfibers.

**Figure 5 polymers-14-01000-f005:**
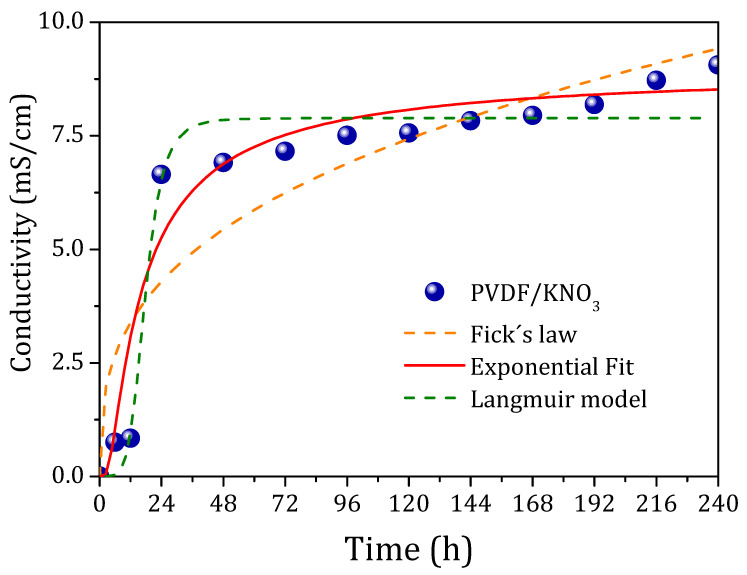
Water conductivity curve as a function of time for PVDF/KNO_3_ sub-microfiber release kinetic study.

## Data Availability

Data available on request.

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
