# Peer review of "PVDF/KNO3 Composite Sub-Microfibers Produced by Solution Blow Spinning as a Hydrophobic Matrix for Fertilizer Delivery System"

_polymers, 2022, doi:10.3390/polym14051000_

Round 1

Reviewer 1 Report

In this manuscript, PVDF/KNO3 composite fiber for fertilizer delivery system was manufactured using solution blow spinning method. Since the sub-microfibers shown in the FE-SEM image contain countless beads, it is difficult to say that the fibers are well manufactured. It is considered that it would have been better to use PVDF-HFP with better processability than PVDF. Additional disusions are required to measure delivery performance more accurately, such as changes in weight and volume of fibers over time. In addition, it is considered that UTM and TGA analyzes that can evaluate the physical properties of polymers are necessary. In addition, it is also necessary to investigate the correlation between the crystal structure of the PVDF composite used and the delivery performance.

Author Response

Dear reviewer, thank you for the comments. Pelase see the attachment with our responses.

with regards

Reviewer 2 Report

Comments to the Authors:

  1. The abstract of the manuscript is well written to present the importance of the study, but still wordy and lacks to mention the most important results obtained!
  2. There are some typo errors in the manuscript; for example Section 2.2, Page 3 Line 98 " reference for describe eache sample. " line 117 " by using 116 diferents solution parameters " please check entire English language of the manuscript.
  3. Page 4, line 138, "As a result, we can observe a high concentration of beads formation in the mat, Figure 3(b)". It is better to highlight the beads for example on the Figure!!
  4. The authors should discuss the results of FTIR first, therefore I suggest move Figure 4 before Figure 2.
  5. No clear information on the measurement of the contact angle of the nanofibers in characterizations section!!! How the authors measured the contact angle during 80 s. The following sentences on the CA should be move to the characterization section and the authors should present deep explanation and discussion on CA! Why CA constant within 80 s for all samples.
  6. The water conductivity during the discussion of Figure 5 was not used!!!!
  7. English language must be edit by native person

Author Response

Dear reviewer thank you for the comments. Please see the attachment with our responses.

with regards

Round 2

Reviewer 2 Report

Dear Editor

I would like to inform you that the authors answer all our comments therefore now it is suitable for publication in Polymers

Best Regards

Author Response

Thank you for considering our manuscript for publication